# Metal Exposure-Related Welder’s Pneumoconiosis and Lung Function: A Cross-Sectional Study in a Container Factory of China

**DOI:** 10.3390/ijerph192416809

**Published:** 2022-12-14

**Authors:** Siyun Zhou, Yue Wang, Chen Yu, Chunguang Ding, Jiayu He, Yi Liu, Huanqiang Wang, Chunhui Ni

**Affiliations:** 1Department of Occupational Medical and Environmental Health, Key Laboratory of Modern Toxicology of Ministry of Education, School of Public Health, Nanjing Medical University, Nanjing 211166, China; 2Department of Occupational Respiratory Disease, National Institute of Occupational Health and Poison Control, Chinese Center for Disease Control and Prevention, Beijing 100050, China; 3National Center for Occupational Safety and Health, Beijing 102300, China; 4Gusu School, Nanjing Medical University, Nanjing 211166, China

**Keywords:** metal exposure, welder’s pneumoconiosis, lung function, working environment, association analysis

## Abstract

Long-term inhalation of welding fume at high exposure can cause welder’s pneumoconiosis, and metals in welding dust are associated with respiratory dysfunction. This cross-sectional study, which contains 384 Chinese male workers who were or had been working in a container factory, aimed to assess the potential risk of haemal and urinary metal content in welder’s pneumoconiosis. Further, we investigated their effects on lung function parameters. Metal content and lung function were measured using inductively coupled plasma–mass spectrometry (ICP-MS) and spirometer, respectively. The concentration and metal content of respirable dust as well as total dust were collected at this container factory. Lung function of cases with welder’s pneumoconiosis was significantly worse, as indicated by lower values of FVC, FVC% predicted, FEV1, FEV1% predicted, MEF25% predicted, and MMEF% predicted (*p* < 0.05). Results of logistic regression models showed that haemal Cr and Zn were risk factors of welder’s pneumoconiosis (OR = 4.98, 95%CI: 1.73–21.20, *p* = 0.009 for Cr; OR = 5.23, 95%CI: 1.56–41.08, *p* = 0.033 for Zn) after adjusted with age, BMI, working years, welding dust exposure years, and smoking status. Multiple linear regression models showed that several metals (haemal Cd and Pb; urinary Cd and Fe) were significantly associated with different lung function indices in the welder’s pneumoconiosis group. Compared to non-welders, welders were exposed to considerably higher levels of respirable dust, total dust, and six kinds of metals (*p* < 0.05). In conclusion, haemal Cr and Zn are positively related to welder’s pneumoconiosis. Meanwhile, Cd and Pb might worsen lung function in welder’s pneumoconiosis.

## 1. Introduction

Electric welding is widely applied in the modern industry involving an estimated 11 million welders worldwide [1]. Welding fume is the main hazard factor to welder’s health. Ample evidence has indicated the association between welding fume and adverse physiological changes or multiple disease outcomes [2,3,4,5]. Welding fume has been classified as carcinogenic to humans (Group 1) by the International Agency for Research on Cancer (IARC) since 2017 [6]. The main carcinogenic components of welding fume are considered to be respirable particles of 20–1000 nm [7]. During the welding process, complex aerosols (0.005–20 µm in size) are generated from the consumption of metal electrodes or wires under extremely high temperatures (>2000 °C) at the weld area [8]. Then, those small particles are carried by thermal updrafts and dispersed throughout the workplace [9]. Particles with different sizes exhibit distinct aerodynamic behaviours. Generally, particles of <2.5 µm can cross the alveolar–capillary barrier, traveling to other organs within the body [10].

Metal particles are enriched in welding dust, and their health hazards have been extensively reported in the literature—the hypothetical mechanisms of metal fume toxicity centre on oxidative stress and inflammatory reaction [11,12]. Inhaled metals could initiate inflammatory responses and cause an imbalance between the production and detoxification of reactive oxygen species (ROS) [13]. A metabolomics study showed evidence that metabolite changes during welding fume exposure were closely associated with systemic inflammation [14]. In addition, several metals in the welding fume (chromium (VI), lead, cadmium, manganese, and some nickel and cobalt oxides) are also recognized as carcinogens [15]. Metal exposure is closely related to changes in lung function. For example, high haemal and urinary Pb levels were associated with decreased lung function and increased risks of respiratory diseases [16,17].

Welder’s pneumoconiosis is caused by long-term inhalation of the welding fume at high exposures, which progresses slowly after being isolated from environmental exposure with continuous and relatively stable performance. It is characterized by diffuse fibrosis in lung tissues and has been listed in the catalogue of statutory occupational diseases in China since 1987 [18]. Welder’s pneumoconiosis cases have lung function abnormalities, but no special treatment is generally required. In this study, we recruited 21 cases with welder’s pneumoconiosis, 69 non-welders, and 294 welders. We also tested participants’ lung functions. The inductively coupled plasma–mass spectrometry (ICP-MS) was applied to determine the metal content in whole blood and urine samples, as well as dust samples collected in a container factory. We aimed to investigate if haemal and urinary metal content were associated with welder’s pneumoconiosis and then explored their effects on lung function parameters.

## 2. Materials and Methods

### 2.1. Population and Study Design

A total of 21 cases with welder’s pneumoconiosis, 69 non-welders, and 294 welders working at a container factory in Qingdao, China, were enrolled in our cross-sectional study. The cases of welder’s pneumoconiosis were diagnosed seven years before this study according to the diagnostic criteria of pneumoconiosis (GBZ70-2009) [19]. All the cases worked as a welder and have left welding jobs after diagnosis but are still working at this factory; furthermore, they had not received any treatment, and no other pneumoconiosis cases were newly found in recent years. The stages of cases (in the first and the last diagnosis) are listed in Appendix A. All cases were still suffering from welder’s pneumoconiosis at the time of investigation. Non-welders in the present analysis referred to workers who do not have direct contact with welding fume, such as auxiliary workers and security guards. All welders met the recruitment criteria as follows: full-time workers; adults (>18 years); qualified to work in a confined space; without medication use, and self-reported history of chronic lung diseases. Our study was carried out in conjunction with the annual occupational health check-up of the enterprise which was completed within one week. Therefore, body fluids were sampled from all participants in the same manner during physical examinations, as well as lung function test. Moreover, 82 welders were recruited to voluntarily participate in the personal sampling process to evaluate welding dust in the air during the corresponding period.

The Ethics Committee approved this research by the National Institute of Occupational Health and Poison Control of the Chinese Center for Disease Control and Prevention, China (No. NIOHP201302). All participants signed the written informed consent. Confidentiality was ensured by not using the names of the workers in any reports.

### 2.2. Questionnaire Investigation

Participants were asked to complete a self-reported questionnaire on age, smoking habits, working years, welding dust exposure years, and respiratory symptoms. The standing height and weight were measured without shoes and were used to calculate the body mass index (BMI).

### 2.3. Spirometry

The spirometer we used to measure lung function was a MasterScreen portable spirometer (Jaeger, Hoechberg, Germany) with a laminar pressure differential flow sensor. The system has a built-in estimated value system, providing a reference standard for lung function of the Chinese population [20]. The specific standard procedures and methods for spirometry can be seen in Appendix A. Two trained physicians performed spirometry according to American Thoracic Society (ATS) standards [21]. Before the test, subjects took a seat to stabilize their breaths. They were required to measure lung function at least 3 times to ensure reliability, and we took the optimal values for analysis. Parameters included in the analysis were forced vital capacity (FVC), forced expiratory volume in one second (FEV1), FEV1/FVC, percent predicted vital capacity (VC% predicted), FVC (FVC% predicted), FEV1 (FEV1% predicted), peak expiratory flow rate (PEF% predicted), maximal expiratory flow at 25%, 50%, 75% of FVC (MEF25% predicted, MEF50% predicted, MEF75% predicted, respectively) and maximal mid-expiratory flow (MMEF% predicted).

### 2.4. Air Sampling

This state-owned container factory has a fixed production process, and the heat source, welding object, and electrode used in electric welding have not changed. In 2014, the ventilation system was improved which reduced the exposure level of welding fume but did not change its contents. After improvement of the ventilation system, we conducted air sampling inside the container factory (Appendix A) according to the specifications of air sampling for hazardous substances monitoring in the workplace (GBZ 159-2004) [22]. The meteorological conditions of the welding workplace were as follows: ambient temperature, 20.8–21.6 °C; air pressure, 101.3 KPa; relative humidity, 45.6–51.2% RH; wind speed, 0.2–1.1 m/s. Before sampling, detailed information (including workshop, types of sampling and welding, and ventilation situation) were recorded.

For personal sampling [23], 82 welders wore the AirChek2000 portable pumps (SKC, Eighty Four, PA, USA) when working for respirable dust testing. The pre-separators (SKC, USA) were installed in front of the 30 mm sampling heads (Yilian, Suzhou, China), and microporous filter membranes (Yousheng, Beijing, China) were used to collect dust with aerodynamic diameter below 7.07 μm (defined as respirable dust). During sampling, the Air Chek2000 portable pumps were hung on the belt around the worker’s waist, and the sampling heads were fixed to the worker’s collar (near the worker’s breathing zone). For area sampling [24], the Gilian Gilair pumps (Sensidyne, St. Petersburg, FL, USA) were placed as close as possible to workers (near the breathing zone but would not affect their work) at 20 representative working sites for total dust testing. The sampling points were located in the downwind direction of the work site, away from the exhaust port and the place where eddy current may generate. Gilibrator2 soap film flow calibrators (Sensidyne, USA) were used for flow correction before and after field sampling. Another 10 environmental samples were collected. The sampling flow of respirable and total dust was 2 and 5 L/min, correspondingly. Glass fibre and mixed cellulose ester (MCE) filter membranes were used for dust concentration calculation and further metals analysis, respectively. All filters were kept in a desiccator to eliminate humidity. For gravimetric analysis, glass fibre filters were pre- and post-weighed on a microbalance in temperature and humidity-controlled conditions.

Dust collected by the MCE filter membrane was used for metal element analysis. Referring to the National Institute for Occupational Safety and Health (NIOSH) method 7303 [25] and the determination of toxic substances in workplace air (GBZ/T 300.33-2017) [26], the membrane was dissolved in 2.5 mL nitric acid and 1 mL hydrogen peroxide and then was digested by microwave electric heating digestion apparatus.

### 2.5. Whole Blood and Urine Samples Collection

Whole blood and urine samples were collected and transported according to the general principles of the biological monitoring method in the occupational population (GBZ/T 295-2017) [27]. For pre-treatment, whole blood samples were diluted by Triton X-100-HNO_3_, while urine samples were diluted by HNO_3_.

### 2.6. Metals Analysis

After sample pre-treatment, we used inductively coupled plasma optical emission spectrometry (5100 ICP-OES, Agilent, Santa Clara, CA, USA) and inductively coupled plasma–mass spectrometry (ICP-MS, NexIon 350D, PerkinElmer, Waltham, MA, USA) to analyse the concentrations of chromium (Cr), cuprum (Cu), iron (Fe), manganese (Mn), nickel (Ni), and zinc (Zn) in collected dust, and measure the content of Cr, Mn, cobalt (Co), Ni, Cu, Zn, molybdenum (Mo), cadmium (Cd), Pb, and Fe in whole blood and urine samples. The limits of detection (LOD) ranged from 0.010 to 0.184 μg/L in laboratory testing (Appendix A). Metal levels of samples that exceeded the LOD were standardized before being recruited into the final analyses. Appendix A lists the chemical composition of welding wires used in the factory.

### 2.7. Statistical Analysis

We first checked the normality of all the data by the Shapiro–Wilk test. Data with skewed distribution or normal distribution were presented as median (interquartile range, IQR) or mean ± standard deviation (SD). The Kruskal–Wallis test and one-way ANOVA test followed by Dunn’s test and LSD *t* test for multiple comparisons were performed to compare skewed distributed and normally distributed continuous variables, respectively, while the Pearson’s chi-squared test and Fisher’s exact test were used for categorical data.

Data on metal levels in blood or urine were log_2_-transformed, and their correlations were tested by Pearson correlation analysis. Logistic regression models were built to assess the associations between haemal and urinary metal concentrations (categories) with welder’s pneumoconiosis, and to estimate the odds ratio (OR) and 95% confidence interval (CI) with adjustment for age, BMI, working years (<1 year, 1–10 years and ≥10 years), welding dust exposure years (<10 years and ≥10 years) and smoking status. In the welder’s pneumoconiosis group, multiple linear regression analysis was conducted to explore the effects of haemal and urinary metal concentrations (categories) on lung function parameters, adjusted for age, BMI, working years, welding dust exposure years, and smoking status.

According to the type of welding and dust exposure level, the welder group was classified into manual welding (n = 149), automatic welding high exposure (n = 29), and automatic welding low exposure (n = 113) group, during which 3 participants engaged in gas tungsten arc welding (GTAW) work were not included. Then, we conducted a subgroup analysis to explore whether there were differences in lung function among people exposed to different levels of welding fume or among people with different types of welding jobs.

The SPSS 26.0 software (SPSS, New York, NY, USA) and R version 4.1.1 (The R Foundation) were used for all statistical analyses in the present study. Data were visualized by R software or GraphPad Prism software (version 6.0). *p* values were two-sided, and *p* < 0.05 was considered statistically significant.

## 3. Results

Compared with welders and non-welders, cases with welder’s pneumoconiosis showed older age, greater BMI values, longer working years, and longer welding dust exposure years (Table 1, *p* < 0.05). The lung function in the case group was significantly worse, as indicated by values of VC% predicted, FVC, FVC% predicted, FEV1, FEV1% predicted and MEF25% predicted (*p* < 0.05), which reflected the tendency of obstructive ventilation dysfunction and impaired small airway function in welder’s pneumoconiosis cases. Moreover, compared to non-welders, welders had significantly lower BMI values, shorter working years, and greater values of MEF25% predicted (Table 1, *p* < 0.05).

A total of 384 whole blood samples and 351 urine samples were collected and used for metals analysis. The concentrations of many haemal and urinary metals in non-welders, welders, and cases with welder’s pneumoconiosis were statistically different (Table 2, *p* < 0.05). Data suggested that the haemal Fe level of welder’s pneumoconiosis group was significantly higher, while urinary Cu and Fe concentrations were statistically lower than both welders and non-welders (*p* < 0.05). Compared to non-welders, the concentrations of Co, Ni, Pb, and Fe in blood, as well as Co, Cd, and Fe in the urine of welders were significantly higher (*p* < 0.05). In contrast, haemal Zn and Mo, as well as urinary Mn, Zn, and Pb of welders were lower than non-welders (*p* < 0.05).

We then explored the potential effect of haemal or urinary metals on welder’s pneumoconiosis. The results of logistic regression models showed that there were no significant associations between urinary metals with welder’s pneumoconiosis risk (Figure 1A). In contrast, haemal Cr and Zn were risk factors of welder’s pneumoconiosis (OR = 4.98, 95%CI: 1.73–21.20, *p* = 0.009 for Cr; OR = 5.23, 95%CI: 1.56–41.08, *p* = 0.033 for Zn) after adjusted for age, BMI, working years, welding dust exposure years, and smoking status (Figure 1B). Next, we conducted multiple linear regression analyses to investigate whether haemal or urine metals were associated with lung function in cases with welder’s pneumoconiosis (Table 3). FEV1, FEV1% predicted, FEV1/FVC, and MEF75% predicted showed negative associations with haemal Cd. In addition, Pb in whole blood was negatively associated with VC% predicted, FVC, FVC% predicted, MEF75% predicted, and MMEF% predicted. A negative association was also observed between FVC and urinary Cd. However, urinary Fe was a protective factor against FEV1/FVC.

Welders were then sub-grouped into manual welding, automatic welding low exposure group, and automatic welding high exposure group according to different types of welding processes and dust exposure levels (excluding three welders engaged in GTAW). The VC% predicted, FVC and FVC% predicted were lower, while the FEV1/FVC were higher in the automatic welding high exposure group (Table 4, *p* < 0.05), which implied a tendency of restrictive ventilation patterns in the lungs of the automatic welding high group population. We further compared the metal levels of welders in each subgroup and the automatic welding high exposure group had higher concentrations of Cr, Co and Ni in whole blood than the other two subgroups (Appendix A).

Our analysis measured the concentration and metal content of welding dust at this container factory. In total, 82 in-plant samples and 6 environmental samples, collected by the personal sampling method, were used for measuring respirable dust (defined as particulate matter with an aerodynamic diameter of less than 7.07 µm). The respirable dust level of in-plant samples (median, 9.06 mg/m^3^) was significantly higher than that of environmental samples (median, 0.16 mg/m^3^), as well as six kinds of metal levels (Fe, Mn, Zn, Ni, Cu, and Cr) in respirable dust (Figure 2A,B, *p* < 0.05). Another 20 in-plant samples and 4 environmental samples collected by the area sampling method were used to determine total dust. The total dust level of in-plant samples (median, 2.52 mg/m^3^) was significantly higher than that of environmental samples (median, 0.29 mg/m^3^), as well as Fe, Mn, Cu, and Zn metal levels in total dust (Figure 2C,D, *p* < 0.05). In addition, 31.6% of calculated 8 h TWA exceeded the permissible concentration–time weighted average (PC-TWA, 4 mg/m^3^) referring to the occupational exposure limits for hazardous agents in the workplace (GBZ 2-2007) [28]. Fe is the most abundant metal in welding dust from this container factory.

## 4. Discussion

In the present study, cases with welder’s pneumoconiosis were diagnosed successively 7 years ago and had been transferred to non-welding jobs in this container factory thereafter, who were still suffering from pneumoconiosis at the time of the investigation. We aimed to analyse the differences in lung function and metal levels among cases with welder’s pneumoconiosis, welders, and non-welders in a cross-sectional study. We measured the metal content of both blood and urine samples and investigated their associations with welder’s pneumoconiosis and lung function parameters. Lung function abnormality was observed in the welder’s pneumoconiosis cases. Our major findings include: (1) Cr and Zn in blood were risk factors of welder’s pneumoconiosis; (2) haemal Cd, urinary Cd, and haemal Pb might be crucial factors that negatively influenced lung function.

In 1936, Doig and McLaughlin first assessed 16 electric arc welders (the main exposure was iron oxide) radiologically and found generalized fine mottling on both lung fields on the chest radiograph of six welders [29]. Subsequently, they followed 15 patients for 9 years and found complete or partial resolution of chest radiograph abnormalities after being isolated from environmental exposure, suggesting that the lung has the ability to recover from the injury caused by iron oxide dust [30]. Similar findings have also been reported. Sun et al. [31] observed that with the extension of divorcing from dust exposure (0–5 years), lung area with small shadows reduced in 8 out of 10 patients. Shi et al. [32] found that 10 out of 39 cases failed to reach the original stages during the 4–6-year follow-up, while the stages of 29 cases remained unchanged. However, it is commonly recognized that after leaving the welding dust environment, welder’s pneumoconiosis still progresses at a relatively slow pace (compared to silicosis); as shown in Appendix A in our study, 6 cases were aggravated, and 15 cases kept at the original stages after leaving welding jobs for 8–10 years. One possible reason to explain the difference in disease development is the overload phenomenon, in which large amounts of fume, the aggregation of iron oxide, and hemosiderin-laden macrophages may exceed the processing capacity of mucociliary movement [33,34]. Moreover, in cases of severe exposure to welding fume, cell injury, hyperplastic repair and abnormal collagen synthesis in lung tissue could lead to irreversible damage to the alveolar structure [35,36]. Epidemiological and experimental research studies have suggested that prolonged inhalation of welding fume suppresses lung defence responses and increases the susceptibility to lung infection [37,38,39,40]. Furthermore, cigarette smoking has a profound effect on the progression and symptom evolution of welder’s pneumoconiosis [41]. Collectively, existing welding dust (containing a variety of hazardous substances that have been deposited in the human body) and constant external irritation (viruses and cigarette smoke) could contribute to the persistence of welder’s pneumoconiosis.

The chronic respiratory risks of welding have been documented [42,43]. A French longitudinal cohort of 21,238 subjects observed a significant accelerated decline in FEV1 (*p* = 0.046) in non-smokers exposed to welding fume after a 5-year follow-up. However, the baseline lung function parameters were higher in welders than in control subjects [44]. Consistent with our results that welders had better lung function than non-welders, a study observed that welders with longer exposures had better lung function at initial cross-sectional analysis, suggesting a “healthy selection” effect [41,45]. Hence, longitudinal studies may be more helpful in digging out the chronic effects of welding fume exposure on lung function [41]. Restrictive ventilation pattern is defined as FVC < 80% predicted, FEV1/FVC ≥ 75% predicted, and FEV1 ≥ 80% predicted value, whereas FEV1/FVC < 75% predicted value, FVC ≥ 80% and FEV1 < 80% predicted value indicate obstructive pattern [43]. Here, we observed a tendency for restrictive ventilation patterns in the lungs of the automatic welding high exposure group population. However, in Pakistani welding workers with exposure longer than nine years, the obstructive ventilatory defect was found and manifested as a significant reduction in FEV1, FEV1/FVC, and PEF relative to controls [42]. The inconsistent defect patterns of lung function among welders [46] are related to the welding type and smoking status. Younes et al. reported that the spirometry pattern in welders involved in flux cored arc welding and shielded metal arc welding was obstructive, and in those involved in gas metal arc welding, it was mixed (obstructive and restrictive pattern) [43]. Aminian et al. found that smoking welders had respiratory disease with a mixed pattern, while non-smokers had a mostly restrictive pattern [47]. Thus, how welding fumes affect lung function and underlying biological mechanisms needs further exploration.

Evidence for the association between Cd exposure and decreased lung function has been well reported. Consistent with our results in the welder’s pneumoconiosis group, a study in a cadmium battery plant showed that high Cd concentration was associated with decreased values for FEV1, PEF, MEF25, MEF50, and MEF75, suggesting a mild airway obstruction [48]. Another cross-sectional study on welders’ lung function found that urinary Cd was negatively associated with FVC and FEV1 [49]. Exposure to other metals could also pose a threat to the respiratory system. For example, a higher level of Pb in blood was associated with decreased values of FEV1/FVC and MMEF in battery and exhaust workers [50]. Fe is an essential metal for haemoglobin synthesis of erythrocytes, oxidation–reduction reactions, and cellular proliferation. The excessive accumulation of Fe has been implicated in the pathogenesis of lung diseases, including asthma and COPD [51]. However, urinary Fe was reported to have positive dose–response associations with FEV1 and FEV1/FVC in the Wuhan cohort [52], which was consistent with our findings.

Cr and Zn in the blood seem to be risk factors for welder’s pneumoconiosis in our study. To the best of our knowledge, there is little evidence of the association between welder’s pneumoconiosis risk with concentrations of haemal Cr and Zn, but exposure to Cr and Zn harms lung function. A cohort study on chromate exposed population suggested that short-term high exposure to Cr was associated with obstructive ventilatory impairment, and long-term exposure further led to restrictive ventilatory impairment [53]. Elevated urinary Zn was associated with lung function reduction, and a dose-dependent association between urinary Zn and restrictive lung disease risk in the general population was previously observed [54]. Thus, whether haemal Cr and Zn increase the risk of welder’s pneumoconiosis through mediating lung function requires further research in cohort studies.

Both personal and area sampling methods were utilized to collect welding dust in this container factory. Area static sampling is mainly suitable for evaluating work environment health status, while personal sampling is superior when assessing individual exposure to dynamic operation since personal samples are collected near the breathing zone [55]. In our study, personal sampling gave higher values than area sampling. Previous research reported that individual exposure measurements were two to ten times greater than values predicted by area sampling, which in our study was about three times [56,57,58]. The comparative performance of these two methods seems to be highly dependent on working conditions, including the welding process, welding current and materials used, ventilation, and the position of the sampler. The metal composition of welding dust mainly depends on the welding process type and materials used (for example, filler material, shield gas composition, and base metal). Fe is the most abundant metal, followed by Zn and Mn in our study. Long-term exposure to iron dust can lead to siderosis with high levels of ferritin depositing in the lung [59,60]. Mn, a known neurotoxicant, would originate from the welding electrode and base metals and could act as a deoxidizing agent to increase the strength of the resultant weld [61]. A population-based study showed that Mn in welding fume and aging might make welders more susceptible to Alzheimer’s disease or related disorders [62]. In comparison, Zn may be derived from anti-rust painting on pipeline surfaces and additives to welding fillers of electrodes or materials. Metal fume fever is likely caused by Zn fume exposure, with typical symptoms such as dyspnoea, fever, and flu-like symptoms [63]. Furthermore, considering that the 8 h TWA of total dust from 31.6% area samples exceeded the occupational exposure limit in this container factory, effective protective measures are necessary to be taken.

## 5. Conclusions

Our present findings suggest that haemal Cr and Zn are risk factors of welder’s pneumoconiosis. In addition, Cd and Pb might be harmful to lung function. Further research on the association among haematuric metals, lung function, and welder’s pneumoconiosis is necessary. More attention should be paid to metal exposure-related lung health in occupational populations.

## 6. Limitations

There are certain limitations in the study. First, the sample size is relatively small. Second, we failed to match welding dust data to the individual participant due to data limitations. Third, smoking habits were based on self-reported data, some of which are missing. Fourth, obvious bias exists in our cross-sectional study, and longitudinal studies should be considered in future studies. Finally, other factors also contribute considerably to metal contents in body fluids; atmospheric and household particulate matter, soil environment, diet, and cigarette smoke are common sources of metal exposure. Thus, the interpretation of our results should be cautious, and further studies are needed to verify our findings.

## Figures and Tables

**Figure 1 ijerph-19-16809-f001:**
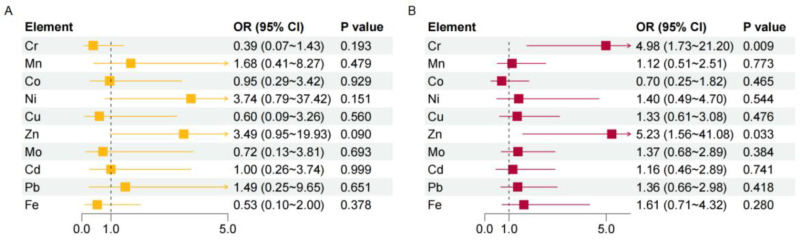
Analysis of the risk of pneumoconiosis for different metals. The associations between urinary (**A**) or haemal (**B**) metals with welder’s pneumoconiosis risk were estimated by logistic regression models with adjustment for age, BMI, working years, welding dust exposure years, and smoking status. *p* < 0.05 indicates statistical significance. OR, odds ratio; CI, confidence interval.

**Figure 2 ijerph-19-16809-f002:**
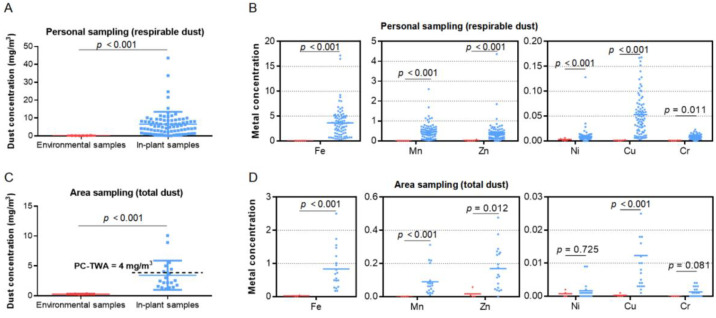
Concentration (mg/m^3^) and content of welding dust. (**A**,**B**) Concentrations of respirable dust and six metals in environmental air samples (red) and in-plant air samples (blue) collected by personal sampling method. (**C**,**D**) Concentrations of total dust and six metals in environmental air samples (red) and in-plant air samples (blue) collected by area sampling method, with 4 mg/m^3^ as the national standard (GBZ 2-2007) for total dust concentration of welding fume. The units for Fe, Mn, Zn, Ni, Cu, and Cr are μg/m^3^. PC-TWA, permissible concentration–time weighted average.

**Table 1 ijerph-19-16809-t001:** Demographic characteristics and lung function parameters of participants.

Variables	Non-welders (n = 69)	Welders (n = 294)	Cases with Welder’s Pneumoconiosis (n = 21)	*p*
Age (year)	29 (7.5)	28 (7)	43 (6.5) ^ab^	<0.001
BMI (kg/m^2^)	23.12 (5.02)	21.67 (3.94) ^a^	25.2 (3.1) ^ab^	<0.001
Working years (year)				
<1	0	12 (4.1%)	0	<0.001
1–10	42 (60.9%)	198 (67.3%)	0	
≥10	27 (39.1%)	84 (28.6%)	21 (100.0%)	
Welding dust exposure years (year)				
<10	-	254 (86.4%)	0	
≥10	-	40 (13.6%)	21 (100.0%)	<0.001
Smoking status				
Current smoker	22 (31.88%)	92 (31.29%)	4 (19.0%)	0.488
Non-current smoker	47 (68.12%)	202 (68.71%)	17 (81.0%)	
Lung function index				
VC% predicted	95.69 ± 11.18	94.54 ±10.74	88.24 ± 13.72 ^ab^	0.023
FVC (L)	4.73 ± 0.63	4.81 ± 0.62	4.01 ± 0.69 ^ab^	<0.001
FVC% predicted	99.82 ± 11.79	98.71 ± 11.23	91.68 ±14.26 ^ab^	<0.001
FEV1 (L)	3.84 ± 0.57	3.96 ± 0.50	3.23 ± 0.56 ^ab^	<0.001
FEV1% predicted	96.34 ± 12.22	96.30 ± 10.57	90.02 ± 14.60 ^ab^	0.043
FEV1/FVC (%)	82.23 (7.23)	82.44 (8.81)	81.37 (8.18)	0.125
PEF% predicted	89.48 (21.14)	86.71 (22.65)	82.40 (26.49)	0.319
MEF75% predicted	90.87 (23.07)	89.28 (26.21)	88.82 (38.93)	0.656
MEF50% predicted	82.18 (30.86)	87.02 (29.76)	76.36 (35.70)	0.190
MEF25% predicted	67.34 (32.77)	77.75 (38.04) ^a^	57.30 (24.19) ^b^	<0.001
MMEF% predicted	82.06 ± 20.84	85.34 ± 20.95	75.08 ± 25.19	0.067

BMI, body mass index; VC, vital capacity; FVC, forced vital capacity; FEV1, forced expiratory volume in one second; PEF, peak expiratory flow rate; MEF75, maximal expiratory flow at 75% of FVC; MEF50, maximal expiratory flow at 50% of FVC; MEF25, maximal expiratory flow at 25% of FVC; MMEF, maximal mid-expiratory flow. Values are mean ± standard deviation (SD), median (interquartile range (IQR)), or number (%). ^a^ Compared with non-welders by Dunn’s test or LSD *t* test. ^b^ Comparison between welders and cases by Dunn’s test or LSD *t* test. *p* < 0.05 indicates statistical significance.

**Table 2 ijerph-19-16809-t002:** Distributions of 10 haemal and urinary metal concentrations in cases with welder’s pneumoconiosis, welders, and non-welders.

	Non-Welders	Welders	Cases with Welder’s Pneumoconiosis	*p*
Element content in blood (log2-transformed)				
Sample size	69	294	21	
Cr	2.85 ± 0.25	2.83 ± 0.46	2.91 ± 0.22	0.653
Mn	4.56 ± 0.36	4.72 ± 0.55	4.59 ± 0.60	0.055
Co	0.30 ± 0.17	0.56 ± 0.30 ^a^	0.38 ± 0.19 ^b^	<0.001
Ni	2.37 ± 0.36	2.61 ± 0.62 ^a^	2.35 ± 0.39	0.002
Cu	9.74 ± 0.15	9.72 ± 0.16	9.70 ± 0.20	0.571
Zn	12.72 ± 0.16	12.51 ± 0.29 ^a^	12.81 ± 0.18 ^b^	<0.001
Mo	1.76 ± 0.41	1.50 ± 0.47 ^a^	1.52 ± 0.58	<0.001
Cd	1.56 ± 0.80	1.71 ± 0.79	1.92 ± 0.74	0.152
Pb	4.86 ± 0.46	5.11 ± 0.49 ^a^	5.15 ± 0.38 ^a^	<0.001
Fe	18.81 ± 0.26	19.05 ± 0.16 ^a^	19.21 ± 0.11 ^ab^	<0.001
Element contents in urine (log2-transformed)				
Number	58	280	13	
Cr	1.20 ± 0.78	1.38 ± 0.74	1.09 ± 0.53	0.105
Mn	2.83 ± 0.87	2.20 ± 0.79 ^a^	2.88 ± 0.63 ^b^	<0.001
Co	0.38 ± 0.20	0.55 ± 0.34 ^a^	0.29 ± 0.33 ^b^	<0.001
Ni	2.93 ± 0.92	2.97 ± 0.77	3.30 ± 0.52	0.294
Cu	7.16 ± 0.59	6.99 ± 0.62	6.47 ± 0.47 ^ab^	0.001
Zn	8.99 ± 0.97	8.60 ± 0.97 ^a^	9.56 ± 0.78 ^b^	<0.001
Mo	5.81 ± 0.93	5.97 ± 1.01	5.15 ± 1.04 ^b^	0.012
Cd	0.72 ± 0.57	0.87 ± 0.38 ^a^	1.12 ± 0.68 ^a^	0.005
Pb	1.69 ± 0.87	1.20 ± 0.72 ^a^	1.93 ± 0.55 ^b^	<0.001
Fe	8.28 ± 0.97	8.69 ± 1.05 ^a^	7.41 ± 1.29 ^ab^	<0.001

Values were log2-transformed and presented as mean ± SD. The units for urinary Fe, Mn, Zn, Ni, Cu, and Cr are μg/L. The units for haemal Mn, Zn, Ni, Cu, and Cr are μg/L, while the unit for haemal Fe is mg/L. ^a^ Compared with non-welders by LSD t test. ^b^ Comparison between welders and cases by LSD *t* test. *p* < 0.05 indicates statistical significance.

**Table 3 ijerph-19-16809-t003:** The association between metal levels and lung function in cases with welder’s pneumoconiosis.

	VC% Predicted		FEV1		FEV1% Predicted		FVC	
	β (95% CI)	*p*	β (95% CI)	*p*	β (95% CI)	*p*	β (95% CI)	*p*
Blood_Cd	1.41	0.716	−3.91	0.028	−4.62	0.036	0.11	0.561
	(−6.05 to 8.87)		(−7.09 to −0.74)		(−8.57 to −0.68)		(−0.26 to 0.48)	
Blood_Pb	−14.66	0.036	−1.57	0.672	−1.97	0.665	−0.71	0.044
	(−27.22 to −2.11)		(−8.69 to 5.55)		(−10.7 to 6.77)		(−1.34 to −0.07)	
Urine_Cd	−9.00	0.118	−0.40	0.102	−8.50	0.228	−0.51	0.044
	(−19.40 to 1.40)		(−0.83 to 0.04)		(−21.53 to 4.53)		(−0.95 to −0.07)	
Urine_Fe	−2.58	0.417	0.02	0.872	−0.60	0.878	−0.08	0.567
	(−8.59 to 3.42)		(−0.24 to 0.28)		(−7.99 to 6.80)		(−0.36 to 0.20)	
	**FVC% Predicted**		**FEV1/FVC**		**MEF75% Predicted**		**MMEF% predicted**	
	**β (95% CI)**	** *p* **	**β (95% CI)**	** *p* **	**β (95% CI)**	** *p* **	**β (95% CI)**	** *p* **
Blood_Cd	1.46	0.711	−3.91	0.028	−12.62	0.031	−9.57	0.047
	(−6.14 to 9.06)		(−7.09 to −0.74)		(−23.08 to −2.16)		(−18.29 to −0.84)	
Blood_Pb	−14.67	0.04	−1.57	0.672	−28.43	0.009	−21.19	0.02
	(−27.54 to −1.80)		(−8.69 to 5.55)		(−47.29 to −9.56)		(−37.24 to −5.13)	
Urine_Cd	−9.34	0.118	0.1	0.959	−5.96	0.537	−4.05	0.613
	(−20.13 to 1.45)		(−3.78 to 3.98)		(−24.28 to 12.35)		(−19.31 to 11.20)	
Urine_Fe	−2.7	0.414	2.12	0.027	3.76	0.460	−1.03	0.809
	(−8.92 to 3.53)		(0.49 to 3.75)		(−5.88 to 13.40)		(−9.20 to 7.14)	

Multiple linear regression analyses were applied to investigate the association after adjustment with age, BMI, working years, welding dust exposure years, and smoking status. *p* < 0.05 indicates statistical significance.

**Table 4 ijerph-19-16809-t004:** Demographic characteristics and lung function parameters of welders in each subgroup.

	Manual Welding (n = 149)	Automatic Welding-Low Exposure (n = 113)	Automatic Welding-High Exposure (n = 29)	*p*
Age(year)	28 (7)	28 (7)	27 (4.5)	0.860
BMI (kg/m^2^)	21.80 (3.74)	21.70 (4.17)	21.01 (2.80)	0.282
Working years (year)				
<1	9 (6.0%)	6 (2.7%)	0 (0%)	0.001
1–10	102 (68.5%)	140 (61.9%)	72 (82.8%)	
≥10	38 (25.5%)	80 (35.4%)	15 (17.2%)	
Welding dust exposure years (year)				
<10	131 (87.9%)	184 (81.4%)	84 (96.6%)	0.001
≥10	18 (12.1%)	42 (18.6%)	3 (3.4%)	
Lung function index				
VC% predicted	95.42 (12.73)	92.78 (13.88) ^a^	90.12 (14.90) ^ab^	0.016
FVC (L)	4.89 (0.67)	4.71 (0.90) ^a^	4.41 (0.85) ^ab^	0.004
FVC% predicted	99.77 (13.37)	97.03 (15.23) ^a^	94.18 (15.75) ^ab^	0.013
FEV1 (L)	3.98 ± 0.46	3.94 ± 0.56	3.95 ± 0.52	0.837
FEV1% predicted	96.65 ± 9.96	95.52 ± 11.38	97.68 ± 10.75	0.530
FEV1/FVC (%)	81.50 ± 7.11	83.11 ± 6.48 ^a^	86.17 ± 5.79 ^ab^	0.002
PEF% predicted	86.08 (26.96)	88.61 (21.01)	83.55 (20.82)	0.539
MEF75% predicted	89.24 (26.06)	86.44 (29.15)	93.83 (21.91)	0.431
MEF50% predicted	86.12 (31.01)	87.80 (29.74)	92.07 (23.68)	0.257
MEF25% predicted	73.73 (36.37)	78.74 (38.94)	87.32 (48.58)	0.066
MMEF% predicted	83.90 ± 21.17	85.68 ± 21.04	90.93 ± 18.64	0.246

Values are mean ± SD, median (IQR), or number (%). ^a^ Compared with manual welding by Dunn’s test or LSD *t* test. ^b^ Comparison between automatic welding low exposure and automatic welding high exposure group by Dunn’s test or LSD *t* test. *p* < 0.05 indicates statistical significance.

## Data Availability

Not applicable.

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
