# Peer review of "Metal Exposure-Related Welder’s Pneumoconiosis and Lung Function: A Cross-Sectional Study in a Container Factory of China"

_ijerph, 2022, doi:10.3390/ijerph192416809_

Round 1
Reviewer 1 Report
In the present manuscript the Authors evaluate the long-term effects of the exposure to welding fumes in Chinese workers: 21 affected by pneumoconiosis, 294 workers directly exposed, and 69 not exposed. Environmental particulate matter sampling was performed and – on 82 workers – personal monitoring was also done. Pulmonary function tests were performed by means of a portable spirometer. Metal content in sampled air and metal concentrations in blood and urine was performed. They found that blood content in Cr and Zn was a risk factor for pneumoconiosis and that some metals were associated with poorer lung function in workers’ pneumoconiosis.
The manuscript is interesting and well written, unfortunately I find it suffers from some methodological problems.
Mainly, Authors declare that this is a cross-sectional study. Nevertheless, the sample group relevant to subjects affected by pneumoconiosis left the working activity after the diagnosis (i.e., 7 years before). Thus, if I well understand, the Authors are trying to relate the development of a disease diagnosed 7 years before (and possibly due to an exposure occurred in the previous years) to analyses (metals in biologic fluids and in the environment) performed long after. If so, this is not reasonable, please explain.
As concerns welding dust collection, in Results section (and not in Materials and Methods, why?) the “respirable dust” is defined as particulate matter with an aerodynamic diameter of less than 7.07μm. But I cannot see any mention of impactors with a selected cut-off for both personal and area sampling. Were AirChek2000 portable pumps able to sample respirable fraction only? If so, how? For area sampling, it is clearly stated that a total dust sampling was performed.
Minor concerns
Materials and Methods, section 2.1 Population and study design, row 79: please, specify when the “cases” were sampled
Materials and Methods, section 2.3 Spirometry, row 96: Please specify which type of spirometer was used (pneumotachograph?) and the relevant adopted calibration procedures
Materials and Methods, section 2.3 Spirometry, row 100: You use the FEV1/FVC% and not FEV1/FVC ratio
Materials and Methods, section 2.3 Spirometry, rows 100-104: Which prediction equations were used?
Materials and Methods, section 2.4 Air sampling, rows 112-113: “…respirable dust…” please add here “defined as particulate 263 matter with an aerodynamic diameter of less than 7.07μm”
Table 2: Why did you log transform data instead of presenting them as medians and IQR?
Table 3: Were metal blood levels put in the same multiple model or in separate models (one for each metal)?
Discussion, rows 287-288: I would not consider finding lung function abnormalities in pneumoconiosis as a "major finding"
Reviewer 2 Report
This manuscript conducts a correlation study between blood and urine metal content and welders' pneumoconiosis to investigate its effect on lung function parameters. The manuscript is interesting, but there are some research design problems:
The 21 welder pneumoconiosis cases studied by the authors were diagnosed seven years ago, but the body fluids and environmental samples taken from the confirmed cases were not from seven years ago but now. I think this is unreasonable. Please explain.
Whether the treatment was the same and the degree of pneumoconiosis was the same in 21 patients with welder's pneumoconiosis within 7 years of diagnosis, There is no relevant information in the article please add relevant information. I suggest that the 21 patients with welder's pneumoconiosis be graded for pneumoconiosis and graded to analyze the effect of metal changes in body fluids on the pneumoconiosis condition.
Minor concerns:
1. It is suggested that a sketch of sampling sites be added to method 2.4.
2. Metal analysis why are the metal elements detected in the dust (CR, Cu, Fe, Mn, Ni, and Zn) and blood and urine samples (CR, Mn, Co, Ni, Cu, Zn, Mo, Cd, Pb, and Fe) different.
3. Why not choose the linear mixed-effects regression model but choose the logistic regression model?
4. The results are described confusingly, please write in a uniform style, it is recommended that these be presented in chart order, for example, lines 172-171 and lines 226-228 are describing table 1.
5. Reasons why correlation analysis between metals in blood and urine was done (Figure 1), suggestion removed.
6. The description in the results should not be Table 1 describes ****** (lines 170-171), Figure 1 presents ****** (line 191), and Table 2 shows ***** (line 192), which should be in the Figure/Table legends. The description of the results should be followed by the source of the figure or table.
7. Discussion is not required in the description of results and is not recommended for citation (line 226).
8. Figure 2 should be titled Analysis of the risk of pneumoconiosis in welders for different metals.
9. Figure 2 should be Figure 2 B (line 217), and the text should be described with Figure 2 A even though there is no significant difference.
10. Cr and Zn in blood as risk factors for welder's pneumoconiosis and why Cr and Pb were selected to see if they are related to lung function?
11.Why is urine Cr and Fe selected to observe whether they are related to lung function?
12. (p < 0.05, Table 4) should be changed to (Table 4, p < 0.05) in line 242, please check in full and harmonize.
13. Please do not describe in the results again describe the experimental method, for example, lines 262-264.
14. Please do not include speculation in the description of the results, for example, “However, urinary Fe seemed to be a protective factor against FEV1/FVC” in lines 223
15. “Demographic characteristics and Lung function parameters of welders in each subgroup” are recommended to delete or further analyze the relationship between the metallic elements of hematuria and lung function in the welders in each subgroup.
16. Whether there is a link between the number of metals in personal and area samples and the number of metals in blood and urine.
Round 2
Reviewer 1 Report
In their point-by-point response to reviewers, the Authors write “Our purpose was … to analyze the differences of lung function and metal levels among cases with welder's pneumoconiosis, welders and non-welders at the time of the investigation”. Among the Authors’ aims, the comparison of lung function between welders and non-welders at the time of the investigation is the only one that presents a rationale. In fact, it is obvious that workers affected by pneumoconiosis have a poorer lung function with respect workers still working at the time of the investigation without any evidence of lung disease. As concerns the comparison of metal levels among cases with welders with previously diagnosed pneumoconiosis, welders and non-welders at the time of the investigation, again, I think that it is unrealistic to compare workers who developed the disease 7 years before and changed their work based on analyses performed 7 years after.
Author Response
We appreciate the evaluation of the manuscript by Reviewer 1, and we understood the concerns of reviewer 1. We have passed an explanation of these concerns back to the editor. Here, we would like to make a short explanation regarding the question of reviewer 1.
In our study, cases with welder's pneumoconiosis were diagnosed 7 years before and they have been transferred to non-welding jobs thereafter in the same factory. All workers underwent annual occupational physical examinations. All cases were and are still suffering from pneumoconiosis at the time of the investigation and even now since welder's pneumoconiosis is a kind of chronic and incurable disease that progresses slowly after diagnosis with continuous and relatively stable performance.
We understood the concerns of reviewer 1. So we described and compared the metal levels among cases, welders and non-welders at the time of the investigation simply. On one hand, we found that “haemal Cr and Zn were risk factors of welder’s pneumoconiosis” (line 236-241). On the other hand, the concentrations of haemal Cr and Zn in cases were the highest (Table 2, line 227-229) even though that cases developed the disease and had changed their work 7 years before, manifesting the crucial effect of haemal Cr and Zn on welder’s pneumoconiosis. Next, we focused on exploring the association of metal levels and lung function only in the population of welder's pneumoconiosis cases. Therefore, we are sure this study design is reasonable.
Hopefully, this explanation will address your concern.
Reviewer 2 Report
The author explains my question very well and suggests publishing.
Author Response
We appreciate the evaluation of the manuscript by Reviewer 2, and we are grateful for your suggestions and recognition of our work, which are useful not only for the improvement of the quality of the manuscript but also for our future studies.